# Chemical Composition of Volatile and Extractive Organic Compounds in the Inflorescence Litter of Five Species of Woody Plants

**DOI:** 10.3390/plants13131829

**Published:** 2024-07-03

**Authors:** Valery A. Isidorov, Jolanta Masłowiecka

**Affiliations:** Institute of Forest Sciences, Białystok University of Technology, 15-351 Białystok, Poland; j.maslowiecka@pb.edu.pl

**Keywords:** inflorescences of woody plants, spring fall, decomposition, emission of VOCs, chemical composition

## Abstract

The decomposition of plant litter, most of which is found in forests, is an important element of the global carbon cycle, as a result of which carbon enters the atmosphere in the form of not only CO_2_ but also volatile organic compounds (VOCs). Although the formation of litter is associated with autumn cooling, in the spring, there is a very intense fall of faded inflorescences of woody plants. This study examined the chemical composition of the litter and VOCs emitted from decaying inflorescences of four species of forest-forming trees: silver birch, European hornbeam, black alder and aspen. All litter emissions consisted of 291 VOCs, mainly terpenes actively participating in atmospheric processes. The detection of a number of typical mushroom metabolites, such as 1-octen-3-ol, known as “mushroom alcohol”, and alkyl sulphides, suggests that inflorescence-derived VOCs are a mixture of components of plant and microbial origin. In methanol extracts of the fallen inflorescences of all types, 263 organic compounds were identified, the majority of which were related to carbohydrates. Their share in the extracts was 72–76%. In general, the composition of the extractive compounds indicates the easy availability of this material for assimilation by various types of destructors.

## 1. Introduction

The formation and further decomposition of plant litter encompass the main element of the natural cycle of carbon and nutrients in the biosphere, including in forest ecosystems. In the process of litter decomposition, controlled by a number of insufficiently studied physical and biological factors, various volatile organic compounds (VOCs) enter the atmosphere [1]. As they are photochemically active, they have a significant effect on chemical processes in the atmospheric boundary layer. Although it has been established that the main biogenic source of VOCs is the living vegetation on the continents [2,3,4,5], in recent decades, increasing attention has been paid to other previously unexplored sources, including plant litter [6,7,8,9,10,11,12,13,14,15,16,17,18,19,20,21,22]. The motivation to study the VOCs produced in leaf litter was, on the one hand, the recognition of their important role as key participants in atmospheric chemical processes and, on the other hand, the high degree of uncertainty in existing estimates of biogenic emissions [23,24]. This uncertainty may be explained by the presence of additional previously unaccounted for sources of biogenic VOCs. The determination of the total reactivity of the hydroxyl radical in boreal forests clearly indicates the existence of a previously unaccounted for source of VOCs active in photochemical processes [25]. This makes it necessary to characterise all natural sources of reactive VOCs [26].

The ecological significance of studying forest plant litter (chemical composition and the dynamics of its changes during the decomposition process) is also associated with the fact that it serves as a habitat and a food resource for a variety of organisms, including many species of saprotrophic invertebrates. These organisms play an active role in the decomposition of litter [27,28,29,30,31], and their activity is also accompanied by the emission of VOCs contained in dead plant material, as well as the release of their own volatile components into the gas phase [1].

In temperate and boreal climate zones, the formation of litter is associated with the fall of leaves in autumn during the onset of cold weather. Therefore, the exchange of trace gases between the forest floor and the atmosphere during the summer–autumn transition period has received much attention [32,33,34,35,36]. However, in forest ecosystems, plant litter is not only made up of dead leaves, and its formation is not limited to the autumn period. In the spring and in the case of some tree species, even before foliage blooms, flowering begins, during and at the end of which part of the inflorescences dies and falls to the ground. The composition of VOCs released by the living flowers of woody plants has received much attention not only because of their involvement in atmospheric processes but also because of their role in interspecies communication. The smell of a flower is specific to each plant species and can be used by pollinators as a signal to locate and recognise the source of nectar or pollen. A recent review [37] quantitatively summarised data on the relative composition and release rates of VOCs in the floral fragrances of 305 plant species from 66 families. These data indicate that the main components of flower-derived VOCs are terpene compounds and benzenoids, which play an important role in atmospheric chemistry. However, until now, there is yet to be a discussion on the chemical composition of faded inflorescences and their role in the processes occurring under the forest canopy.

We hypothesise that the release of volatiles from spent inflorescences continues even after they have fallen, and, thus, this process serves as a source of highly reactive VOCs during the spring–summer transition period. In addition, it can be assumed that the chemical composition of fallen inflorescences makes them an attractive food resource for saprotrophic organisms. In accordance with this, the purpose of this work was to study the chemical composition of the volatile emissions of the fallen inflorescences of forest-forming woody plants in the mid-latitude and boreal zones of the northern hemisphere and to determine the composition of the extractive compounds contained in them.

## 2. Experimental Section

### 2.1. Chemicals

Pyridine, a ready-to-use mixture of bis(trimethylsilyl)trifluoroacetamide with 1% trimethylchlorosilane (BSTFA/TMCS) and a C_8_–C_40_ *n*-alkane calibration standard were purchased from Sigma-Aldrich (Poznan, Poland). Pure p.a. methanol and anhydrous magnesium sulphate (MgSO_4_) were acquired from Chempure (Piekary Śląskie, Poland). All solvents were used without any purification. 

### 2.2. Plant Material and Techniques for Field Experiments

This study examined the composition of organic compounds released into the gas phase from the inflorescence litter of five species of deciduous trees: silver birch (*Betula pendula*), European hornbeam (*Carpinus betulus*), Norway maple (*Acer platanoides*), aspen (*Populus tremula*) and crack willow (*Salix fragilis*). All these types of woody plants are characteristic of forests not only in Poland but also in the entire temperate and boreal zone of the European continent. Freshly fallen inflorescences of all plant species were hand-collected from the Las Zwierzyniecki Nature Reserve near Bialystok, Poland, in early May 2022. The reserve’s vegetation consists of a stand of trees in which the dominant species is hornbeam, with admixtures of English oak, ash, aspen, maple, silver birch, black alder and various species of willow (*Salix* spp.). In the groundcover layer, we could distinguish the following plants: yellow wood warbler, yellow wood anemone, wood anemone, greater chickweed and common ground elder. 

Some of the collected inflorescences were used in field experiments to study the composition of VOCs during decomposition under natural conditions. For incubation, litter bags (200 × 200 × 20 mm) with a terylene mesh bottom and a mesh size of 1.5 mm were used; each bag contained approximately 10 g of inflorescences. The litter bags were laid out on the natural layer of the previous year’s leaf litter under the trees whose flower litter was studied. The bags were covered on top with a terylene mesh (mesh size 5 mm) to prevent wind drift and foreign materials from having an effect. The duration of this stage of the experiment was 14 days. 

### 2.3. VOC Determination

The determination of the composition of the volatile components of the freshly collected inflorescence litter and that incubated for two weeks was carried out according to a previously described method using solid-phase microextraction combined with gas chromatography with mass spectral detection (HS-SPME/GC-MS) [38,39]. According to the results of the experiments described in the cited papers, which were carried out to determine the composition of VOCs of various natural materials from the available range of sorption fibres, the choice was made to use DVB/CAR/PDMS fibre (Sigma-Aldrich, Poznan, Poland).

The inflorescence litter delivered to the laboratory was placed in a glass container with a volume of 0.25 L, and this was sealed using a lid with a port to introduce the sorption fibre through a silicone membrane. After standing for 0.5 h at room temperature, fibre preconditioned according to the producer’s recommendations was introduced into the gas phase above the leaves. After exposing the fibre for 2 h (every 15–20 min, the contents of the container were shaken), it was injected for 15 min into the injection port of a gas chromatograph HP7890A with a 5975C VL MSD Triple-Axis Detector (Agilent Technologies, Santa Clara, CA, USA). The apparatus was fitted with an HP-5ms capillary column (30 m × 0.25 mm i. d., 0.25 μm film thickness), with electronic pressure control and a split/splitless injector. The latter was operated at 220 °C in splitless mode. The helium flow rate through the column was 1 mL min^−1^ in constant-flow mode. The initial column temperature was 40 °C, and it rose to 220 °C at a rate of 3 °C min^−1^. After integration, the fraction of each component in the total ion current (TIC) was calculated. The precision of the method was studied by three replicate extractions and analyses. The peak areas of the extract components obtained by replicate analyses were used for the calculation of their relative standard deviation (RSD) values. On average, the RSD amounted to 2% for the main peaks (more than 10% of the TIC), 6% for medium peaks (more than 1% of the TIC) and 18% for peaks that amounted to ≤0.5% of the TIC.

To calculate the linear temperature-programmed retention indices (*RI*s) of the analytes, the SPME fibre was inserted for 2–3 s into the headspace of the vial with a mixture of C_6_–C_18_ *n*-alkanes. Their separation was performed under the above conditions. The *RI* values of the separated components were calculated according to the following equation:*RI*^Calc^ = 100[*n* + (*t_x_* − *t_n_*)/(*t*_*n*+1_ − *t_n_*)],
where *t_x_* is the retention time of the analyte, *t_n_* is the retention time of the *n*-alkane eluting directly before the analyte, and *t_n+_*_1_ is the retention time of the *n*-alkane eluting directly after the analyte.

### 2.4. Determination of the Composition of Extractive Substances

A total of 5 g of freshly harvested inflorescences was ground to a size of about 0.5 mm, placed in a conical flask, filled with 25 mL of methanol and placed on a magnetic stirrer. After extraction at room temperature with stirring for half an hour, the solvent was separated by filtration through a paper filter. Extraction with fresh portions of methanol was repeated twice. The combined extracts were dried by adding MgSO_4_ and then evaporated to dryness on a rotary evaporator. Of the resulting material, 5–10 mg was transferred to a 2 mL vial and dissolved in 220 µL dry pyridine, and 80 µL BSTFA/TMCS was added. The reaction mixture was heated for 0.5 h at a temperature of 60 °C to obtain trimethylsilyl (TMS) derivatives.

The GC separation of TMS derivatives and their identification were carried out on the GC-MS apparatus mentioned in Section 2.4, equipped with the same HP-5ms capillary column. The helium flow rate through the column was 1 mL min^−1^. One microliter of the sample was injected with the aid of an HP 7673 autosampler. The injector heated to 280 °C worked in split mode; the split ratio was 10:1. The initial column temperature was 50 °C, rising to 325 °C at 3 °C min^−1^. Electron ionisation mass spectra were obtained at 70 eV of ionisation energy. Detection was performed in full-scan mode from 41 to 600 a. m. u. After integration, the fraction of each component in the total ion current (TIC) was calculated.

The separation of C_8_–C_40_ *n*-alkanes in *n*-hexane was carried out under the above conditions, and the recorded retention times were used to calculate the linear temperature-programmed retention indices (*RIs*) of the separated compounds. 

### 2.5. Component Identification

The separated components of the volatile emissions and the TMS derivatives of the compounds extracted from the fallen inflorescences were identified by their mass spectra using an automatic GC/MS processing system equipped with an National Institute of Standards and Technology NIST 14 electron ionisation mass spectra library. The calculated values of the retention indices were used as an independent analytical parameter. The mass spectrometric identification was considered reliable if its results were confirmed by the calculated *RI* values, that is, if their deviation did not exceed ±10 units from the standard *RI* values published in accessible databases [40,41,42]. If the results of the mass spectrometric identification were not confirmed by the *RI* values due to their absence in the available databases, or if the discrepancy exceeded 10 u.i., the identification was considered tentative.

## 3. Results and Discussion

### 3.1. VOC Composition

The composition of volatile emissions was determined for freshly fallen inflorescences and those exposed to natural conditions for two weeks. The obtained chromatograms of the VOCs in both sets of experiments contained peaks of 291 compounds, whose individual contribution to the total ion current of the chromatograms was not lower than 0.01%. Of these, 246 compounds (84% TIC) were identified by their mass spectra and calculated retention indices; most of the unidentified peaks belonged to minor (0.01–0.05% TIC) VOC components. The chromatograms of the litter of the different species contained 51 to 119 peaks of organic C_1_–C_20_ compounds of various classes. The largest number of peaks was recorded in the chromatograms of hornbeam (*C. betulus*) inflorescences and the smallest in the chromatograms of silver birch (*B. pendula*) inflorescences.

The components identified during the GC-MS analysis were divided into 12 groups, as shown in Table 1, together with the main representatives of each group (only compounds are given whose share in the TIC of at least one of the chromatograms was not less than 0.5%). The complete composition of the volatiles is shown in Appendix A. Although representatives of all 12 VOC groups were present in the volatile emissions of the inflorescence litter of each studied plant species, their individual compositions turned out to be specific. Only 11 identified compounds were common to all samples: acetic acid, toluene, *p*-cymene, α- and β-pinenes, 3-carene, limonene, α-copaene, β-caryophyllene, nonanal and 1-undecene. This kind of compositional specificity was also observed in the case of the fresh and exposed litter of each species. For example, of 146 hornbeam VOCs, only 48 (32%) were recorded in both chromatograms. The lowest similarity (only 16% of “common” components) was observed in the case of maple (*A. platanoides*) inflorescence fall.

Terpene compounds, including 43 monoterpenes and 63 sesquiterpenes, formed the group with the most components in both the “fresh” and “not-fresh” (exposed for two weeks) litter. In the group of terpene compounds, α-pinene, 3-carene, limonene, α-copaene and β-caryophyllene accounted for the largest share. The second largest group (38 components) was formed by carbonyl C_3_–C_12_ compounds (common to all samples, with aliphatic aldehyde nonanal present at the highest concentration). The next largest group (28 compounds) was formed by C_6_–C_16_ alkanes and alkenes, which contributed 1.2–21% to the TIC of the different chromatograms. A significant contribution to the ion current of the chromatograms (from 2.5 to 28% TIC) was also made by 24 aromatic C_7_–C_10_ compounds, which included hydrocarbons and their oxygenated derivatives, such as benzaldehyde, benzyl alcohol and 2-phenylethanol. Aliphatic alcohols, carboxylic acids and furans had a smaller but nevertheless significant contribution.

In all four cases, the chromatograms of the inflorescences exposed for two weeks showed a greater number of peaks than those of the freshly collected inflorescences of the same species (last row of Table 1). Both qualitative and quantitative changes in VOC composition were particularly noticeable in the case of sesquiterpenes. It can be assumed that “new” compounds formed as a result of various kinds of destructive processes, such as the hydrolysis of glycosides. This hypothesis is supported by the fact that, in plant tissues, most terpenoids are represented by various glycosides [43,44].

It is also impossible to exclude the participation of microorganisms–destructors both in the biochemical decomposition of plant glycosides and in the formation of volatile compounds, including terpenoids, as their inherent waste products [45,46]. The participation of litter-degrading microorganisms in the emission of VOCs is supported by the presence of characteristic components such as C_8_ alcohols and carbonyl compounds: unsaturated alcohol 1-octen-3-ol (known as “mushroom alcohol”), ketone 1-octen-3-one and octan-3-one [47]. The S-containing compounds found in the VOCs also had a microbiological origin; dimethyl sulphide and dimethyl disulphide have been found in the volatile secretions of fungal species, such as *Aspergillus versicolor*, *Penicillium commune* and *Phialophora fastigiate*, cultivated on different media [45].

In summary, the volatile emissions from the flower litter of the studied species of woody plants are a blend of low-molecular-weight metabolites of plant and microbial origin. Among them, terpene compounds predominate, unsaturated in their chemical nature. Their lifetime in the atmosphere is limited to a few minutes due to rapid gas-phase reactions with permanent components of the Earth’s atmosphere such as OH and NO_3_ radicals [48]. These processes lead to the rapid formation of toxic photo-oxidants (ozone and organic peroxides), many of which exhibit phytotoxic effects and have a negative impact on plant communities, as well as human health [49]. An increase in the tropospheric ozone concentration can also affect the climate by perturbing the Earth’s radiation budget, as O_3_ is the third-most important greenhouse gas [50]. In addition, relatively low volatile compounds during gas-phase oxidation form products prone to nucleation and secondary aerosol formation, which also affect the radiative budget of the troposphere [51].

According to our observations, this source of VOCs during the transition from spring to summer is relatively short term, occupying a time interval of several weeks. This is due not only to the rapid microbiological decomposition of flower litter, which does not contain stable biopolymers, such as lignocellulose and lignin, but also to the activity of invertebrates for which flower litter serves as food. For example, insects, mainly *Eurydema* sp., were found in the litter bags covered with a mesh with large cells (mesh size of 5 mm). The nutritional value of flower litter for herbivorous insects can be judged by the composition of the extractive substances that it contains.

### 3.2. Composition of Extractive Compounds

The chromatograms of the methanol extracts of the flower litter contained 263 peaks of organic compounds. Of these, 156 compounds (59%) were positively identified. Another 72 peaks were assigned to a specific class of compounds based on mass spectral data (a set of characteristic *m*/*z* values and the overall pattern of mass spectral fragmentation). Most belonged to the group of carbohydrates, totalling many thousands of individual compounds poorly represented in the available spectral and chromatographic databases (this circumstance makes unambiguous identification by GC-MS at the level of a chemical compound almost impossible). A characteristic of this group is the presence in the mass spectra of the TMS derivatives of a set of ions with *m*/*z* 204, 217, 361, 147 and 103 [42].

The identified compounds were roughly divided into seven groups, as shown in Table 2, along with the main representatives of each group. The eighth group consisted of compounds whose mass spectra did not allow them to be reliably assigned to any specific group of organic compounds. The full list of identified components is given in Appendix A. The largest number of peaks (133) was recorded in the chromatogram of the maple inflorescence litter and the smallest number in the case of willow (67 peaks, last line in Table 2). The group with the most extractive compounds and making the greatest contribution to the TIC of all chromatograms was formed by monosaccharides and related compounds, such as sugar alcohols and acids, as well as some amino sugars. Of these, the main ones were glucose (18–21% TIC) and fructose (11–14% TIC), each of which was represented by three different anomers.

The second most important group was formed by di- and trisaccharides, as well as glycosides, which accounted for 14 to 33% of the TIC. This group contained the largest quantities of sucrose, of which the relative contribution to the TIC was 2–13%. The aglycone components of glycosides were phenols (pyrocatechol, hydroquinone and 2-methoxyhydroquinone), phenolcarboxylic acids (salicylic and vanillic acids) and other phenolic compounds, such as sinapyl alcohol and tyrosol (4-hydroxyphenylethanol), as well as many flavonoids. The latter included catechin and *epi*-catechin, kaempferol, naringenin and quercetin. Phenolic compounds were also present in the extracts in a free state, forming the third largest group of extractive components. Their content in the extracts was 4–7% of the TIC. The contribution to the total ion current of the chromatograms of aliphatic C_2_–C_22_ acids was approximately at the same level. In addition, the extracts of the aspen and maple flower litter contained a noticeable amount (7.41 and 2.95%, respectively) of free amino acids, including non-proteinogenic γ-aminobutyric acid. Interestingly, these components were almost completely absent from the flower litter of hornbeam and birch, which belong to the Betulaceae family.

Thus, this study indicates a high content of easily digestible organic compounds in the studied tree plant litter, which makes it an important seasonal resource for various inhabitants of forest soil, both microbes and other organisms of different levels of organisation. These microbial–zoological interactions in turn stimulate the intensification of abiotic processes, such as the physical evaporation of volatiles from degraded litter [1].

## 4. Conclusions

One of the results of this research is the confirmation of the hypothesis that the litter of woody plant inflorescences releases a range of VOCs into the atmosphere, with predominant participation in the emission of terpene compounds. During the spring–summer transition period, this source contributes to the pool of highly reactive components in the air of forests, the scale and significance of which require further study. The importance of this is explained by the participation of litter-derived VOCs in atmospheric processes leading to the formation of secondary pollution affecting the health of people and ecosystems.

A study of the composition of extractive compounds in flower litter showed a high content of easily biodegradable and easily metabolised compounds. This determines their accessibility to communities of microbes and saprotrophs, which play an important role in the life of forests and contribute to the emission of environmentally important VOCs into the atmosphere.

## Figures and Tables

**Table 1 plants-13-01829-t001:** Chemical composition (% of TIC) of VOCs emitted by fallen inflorescences of selected forest-forming deciduous trees of the boreal and mid-altitude zones of Europe. *A*—freshly fallen inflorescences, *B*—inflorescences after two weeks of decomposition in litter bags. 1—*Carpinus betulus*, 2—*Populus tremula*, 3—*Acer platanoides*, 4—*Betula pendula*, 5—*Salix fragilis*.

Compound	*RI^calc^*	1		2		3		4		5	
		*A*	*B*	*A*	*B*	*A*	*B*	*A*	*B*	*A*	*B*
Tricyclene	919	- *	-	0.23	0.58	-	-	-	-	-	-
3-Thujene	925	-	-	trace **	0.30	2.17	-	-	0.39	-	-
α-Pinene	936	1.10	0.06	6.81	0.51	2.82	0.19	4.42	1.31	trace	0.44
Camphene	949	-	-	0.72	0.52	trace	-	trace	0.24	-	0.35
Sabinene	973	-	0.299	0.36	trace	10.97	trace	-	trace	-	0.09
β-Pinene	976	trace	0.28	0.50	0.29	0.72	trace	2.58	0.80	-	0.12
Myrcene	991	-	-	-	-	2.37	-	0.81	1.61	2.22	trace
α-Phellandrene	1005	-	-	-	0.28	-	-	-	20.04	-	15.37
3-Carene	1011	1.50	0.16	1.88	4.26	0.77	0.68	trace	8.88	0.55	0.54
α-Terpinene	1017	-	-	-	-	0.69	-	-	-	trace	-
Limonene	1030	2.19	0.51	0.48	2.26	1.17	0.996	2.41	8.73	0.26	6.74
(*E*)-β-Ocimene	1050	-	0.59	0.56	-	5.49	-	-	-	3.90	-
γ-Terpinene	1059	-	-	-	-	1.19	0.02	-	-	3.90	-
Terpinolene	1085	-	-	-	-	0.50	-	-	-	-	0.38
Linalool	1101	-	0.59	-	-	-	-	-	-	1.69	-
*keto*-Pyranolinalool oxide	1107	0.40	-	-	-	-	2.72	-	-	-	-
(*E*)-Pinocarveol	1137	1.02	1.64	-	-	-	-	6.55	-	-	-
(*E*)-Verbenol	1142	-	-	-	-	-	-	9.31	-	-	-
Camphor	1144	-	-	-	-	-	-	-	0.35	-	0.62
Borneol	1162	-	-	-	-	-	-	1.55	-	-	-
4-Terpineol	1173	-	-	-	-	-	-	2.00	-	trace	-
α-Terpineol	1191	-	-	-	-	-	-	4.32	-	-	-
(*E*)-Carveol	1217	-	-	-	-	-	-	1.16	-	-	-
Citronellol	1231	-	-	-	-	-	-	-	-	0.65	-
Myrtanol	1259	1.30	1.27	-	-	-	-	-	-	-	-
Bornyl acetate	1285	0.29	0.20	trace	0.18	-	trace	8.18	-	trace	0.10
α-Terpenyl acetate	1354	-	-	-	-	-	-	-	2.95	-	-
**Monoterpene compounds**		**18.34**	**10.50**	**11.87**	**10.43**	**31.87**	**16.38**	**62.88**	**35.19**	**11.71**	**26.73**
Unknown sesquiterpene 1	1298	1.30	-	-	-	-	1.05	-	-	-	-
Unknown sesquiterpene 2	1307	2.57	-	-	0.11	-	3.03	-	trace	-	-
Unknown sesquiterpene 3	1318	0.68	-	-	-	-	0.29	-	-	-	-
Unknown sesquiterpene 6	1345	-	-	-	0.38	-	1.54	-	-	-	0.15
α-Cubebene	1350	-	9.91	0.18	1.00	-	-	-	-	-	-
Longicyclene	1367	-	-	-	0.69	-	0.73	-	-	-	-
α-Ylangene	1372	0.65	2.82	trace	0.48	-	4.87	1.20	4.38	trace	0.38
α-Copaene	1376	1.51	6.44	0.80	2.48	trace	11.18	1.14	8.99	0.88	1.10
β-Bourbonene	1388	0.44	0.93	0.45	2.10	-	1.18	-	0.27	0.57	trace
Longifolene	1402	-	-	-	1.02	-	-	10.43	6.27	-	-
β-Caryophyllene	1417	0.28	0.63	0.87	2.77	1.04	2.34	trace	0.72	0.46	0.11
Guaia-6,9-diene	1441	0.74	3.10	-	0.57	-	5.99	-	0.55	0.05	0.30
Aromadendrene	1443	0.37	-	0.35	0.72	-	1.40	-	1.44	-	-
9-*epi*-β-Caryophyllene	1458	-	-	-	-	-	-	0.26	2.69	-	0.10
Alloaromadendrene	1463	-	1.16	-	0.90	-	1.86	-	2.05	-	-
γ-Muurolene	1472	-	0.62	0.23	1.40	-	1.69	-	1.00	-	-
α-Amorphene	1479	-	0.33	-	-	-	0.50	1.55	0.43	-	-
(*E*,*E*)-α-Farnesene	1511	-	1.53	1.20	-	21.67	-	-	-	10.44	-
γ-Cadinene	1516	-	0.36	0.43	1.82	-	0.79	-	0.70	-	-
δ-Cadinene	1524	-	0.41	0.37	2.37	trace	1.20	-	1.11	-	-
Caryophyllene oxide	1581	-	0.14	-	-	-	-	5.04	-	-	-
**Sesquiterpene compounds**		**14.71**	**22.52**	**5.99**	**28.92**	**26.88**	**52.86**	**28.62**	**35.28**	**13.00**	**2.26**
Toluene	763	16.28	0.63	5.63	6.47	3.36	11.80	-	5.57	0.28	33.93
Styrene	890	-	1.04	15.36	24.81	-	1.27	-	-	7.38	0.19
Benzaldehyde	959	-	1.67	0.68	-	-	-	-	-	0.51	trace
Mesitylene	993	-	-	-	1.93	-	-	-	-	4.45	0.69
*p*-Cymene	1023	0.19	0.33	0.37	2.98	0.97	0.94	1.47	3.91	0.02	4.55
Benzyl alcohol	1035	trace	1.20	4.49	-	-	-	-	-	8.04	-
Salicyl aldehyde	1042	-	-	1.04	-	-	-	trace	-	1.12	-
2-Methyxyphenol	1090	-	-	-	-	-	-	-	-	0.51	-
2-Phenyl ethanol	1114	0.33	1.52	0.27	-	trace	-	-	-	3.77	-
*p*-Cymen-8-ol	1183	-	-	-	-	-	-	-	1.56	-	-
Benzyl tiglate	1499	-	-	-	-	-	-	-	-	1.27	-
**Aromatic compounds**		**17.52**	**7.01**	**27.85**	**46.73**	**4.23**	**14.68**	**3.02**	**9.48**	**29.24**	**40.92**
Methanol	-	-	-	3.21	-	-	-	-	-	-	-
Ethanol	-	-	-	-	-	-	-	1.15	trace	-	-
1-Butanol	663	-	-	-	-	1.80	-	-	-	-	-
3-Methylbutanol	725	-	0.15	-	-	-	0.66	-	-	0.16	-
2-Methylbutanol	732	-	-	6.06	-	-	-	-	-	0.64	-
(*Z*)-3-Hexen-1-ol	854	-	-	0.35	-	1.00	-	-	-	0.37	-
1-Hexanol	867	0.25	0.45	0.73	-	0.61	-	-	-	0.43	0.11
2-Heptanol	904	-	-	-	-	0.31	-	-	-	0.86	-
1-Heptanol	971	-	-	-	-	-	-	-	-	1.47	-
3-Ethyl-4-methylpentan-1-ol	1024	-	-	-	-	-	-	-	-	0.76	-
3-Methyloctan-2-ol? ***	1068	-	-	-	-	-	-	-	-	1.87	-
1-Octanol	1071	trace	2.39	-	-	2.27	-	-	-	2.21	-
1-Nonanol	1174	-	0.55	-	-	-	-	-	-	2.90	-
**Aliphatic alcohols**		**18.15**	**4.98**	**14.81**	**0.51**	**6.13**	**1.02**	**1.15**	**4.51**	**8.01**	**0.11**
Acetone	-	-	1.37	0.93	-	-	-	3.93	0.52	0.01	-
2-Butanone	600	-	-	-	-	-	-	-	-	1.03	-
3-Pentanone	701	-	-	2.21	-	0.73	-	-	-	trace	-
Hexanal	801	trace	2.93	trace	-	-	-	-	0.59	-	0.80
(2*E*)-Hexenal (leaf aldehyde)	852	-	2.62	-	-	0.26	-	-	-	0.23	-
Heptanal	903	trace	0.63	-	-	-	-	-	-	-	0.13
3-Octanone	989	4.80	-	-	-	-	-	-	3.96	-	0.23
Octanal	1002	-	1.99	-	-	-	-	-	-	-	-
(*E*,*E*)-3,5-Octadien-2-one	1072	-	-	-	-	-	-	-	-	-	0.64
Nonanal	1106	1.26	9.34	0.57	0.34	trace	0.02	trace	0.57	4.33	1.28
Decanal	1206	0.22	1.27	trace	0.11	-	-	-	-	0.51	0.25
(*E*)-2-Decenal	1262	-	0.82	-	-	-	-	-	-	trace	-
**Aliphatic carbonyls**		**6.28**	**39.30**	**6.67**	**0.11**	**3.80**	**-**	**3.3**	**6.12**	**6.49**	**3.48**
Formic acid	-	-	0.33	-	1.54	-	-	-	-	0.01	-
Acetic acid	661	0.37	1.89	4.00	1.71	2.37	1.58	trace	2.47	0.51	trace
**Aliphatic acids**		**0.37**	**2.60**	**4.25**	**3.25**	**2.37**	**2.97**	**1.91**	**2.47**	**0.52**	**trace**
3-Methylbutyl isopentoate	1107	-	-	-	-	-	-	-	-	1.00	trace
2-Methylbutyl isopentanoate	1110	-	-	-	-	-	-	-	-	1.47	-
(2*Z*)-2-Pentenyl pentanoate	1152	-	-	-	-	-	-	-	-	0.81	-
Isopentyl tiglate	1196	-	-	-	-	-	-	-	-	4.68	-
(*Z*)-3-Hexenyl pentanoate	1234	-	-	-	-	-	-	-	-	2.53	-
Hexyl 2-methylbutanoate	1238	-	-	-	-	-	-	-	-	1.57	0.12
Prenyl tyglate	1245	-	-	-	-	-	-	-	-	2.02	trace
(Z)-3-Hexenyl (*E*)-2-methyl-but-2-enoate	1326	-	-	-	-	-	-	-	-	1.57	-
Hexyl tiglate	1332	-	-	-	-	-	-	-	-	0.50	-
**Aliphatic esters**		**0.64**	**-**	**0.49**	**-**	**-**	**-**	**-**	**-**	**17.64**	**0.12**
2-Methylfuran	606	0.06	-	-	-	-	trace	1.56	0.45	-	-
3-Methylfuran	613	0.59	-	2.11	-	2.37	0.10	trace	0.75	-	7.71
2-Ethylfuran	705	-	0.53	trace	trace	-	-	trace	-	-	4.00
(*E*)-2-(2-Pentenyl)furan	1004	-	-	-	-	-	-	-	-	0.41	-
**Furans**		**3.51**	**1.93**	**5.09**	**trace**	**2.37**	**0.11**	**1.56**	**1.20**	**0.41**	**11.71**
Isobutyl nitrile	623	-	-	-	-	1.91	-	-	-	-	-
2-Methylbutyl nitrile	721	-		0.31	-	3.09	-	-	-	-	-
3-Methylbutyl nitrile	729	-	-	2.44	-	1.70	-	-	-	-	-
**N-Containing compounds**		**-**	**0.07**	**2.75**	**trace**	**6.70**	**-**	**-**	**-**	**-**	**-**
Dimethyldisulphide	742	-	0.25	1.01	trace	-	trace	-	0.17	0.32	0.15
Dimethyltrisulphide	966	-	0.20	0.31	-	-	-	-	-	-	-
**S-Containing compounds**		**-**	**0.45**	**1.32**	**trace**	**-**	**trace**	**-**	**0.17**	**0.32**	**0.15**
(*E*,*Z*)-2,4-Hexadiene	635	-	-	-	-	-	-	-	-	-	8.85
1-Octene	795	0.88	0.14	0.42	-	trace	0.48	-	trace	-	-
1,3-Octadiene	822	1.03	-	-	0.38	-	-	-	0.37	-	-
1-Undecene	1091	5.18	0.46	0.25	1.81	0.24	5.85	trace	trace	0.90	-
*n*-Tridecane	1300	1.89	0.62	0.33	-	-	trace	-	0.40	0.43	-
**Alkane and alkene**		**20.81**	**6.26**	**18.13**	**5.54**	**14.79**	**8.88**	**1.20**	**1.99**	**2.19**	**9.92**
Diethyl ether	-	0.28	-	-	0.60	1.20	-	-	-	-	-
3,7,7-Trimethyl-1,3,5-cyclo-pentetriene	967	-	-	-	2.35	-	-	-	-	-	-
1,3,3-Trimethyl-2-oxabicyc-lo [2.2.2]octan-6-one	1214	-	-	-	-	-	1.83	-	-	-	-
Diterpene C_20_H_32_	1946	-	1.37	-	-	-	-	-	-	-	-
**Other compounds**		**0.28**	**1.37**	**-**	**2.95**	**1.20**	**1.83**	**-**	**-**	**-**	**-**
**NN**		**1.34**	**0.88**	**2.95**	**2.54**	**0.44**	**3.81**	**1.07**	**3.12**	**2.02**	**2.70**
**Peak number**		**74**	**119**	**84**	**92**	**58**	**86**	**51**	**63**	**85**	**66**

* not found; ** below 0.01% of TIC; *** the identification of the corresponding compound is considered preliminary.

**Table 2 plants-13-01829-t002:** Chemical composition of methanol extracts of fallen inflorescences of selected forest-forming deciduous trees. 1—*C. betulus*, 2—*P. tremula*, 3—*A. platanoides*, 4—*B. pendula*, 5—*S. fragilis*.

Compound (TMS Derivative)	*RI^calc^*	Relative Composition (% of TIC)				
		1	2	3	4	5
**Amino acids**		**0.61**	**7.41**	**2.95**	**- ***	**15.21**
Valine, mono-OTMS	1089	-	0.12	trace **	-	0.23
Alanine, di-N,O-TMS	1114	-	0.19	0.14	-	0.38
Valine, di-TMS	1227	-	0.45	0.23	-	2.05
Serine, O,O-di-TMS	1265	-	trace	0.08	-	-
Leucine, N,O-di-TMS	1284	-	0.54	0.12	-	1.12
Proline, di-TMS	1303	-	0.30	0.83	-	0.47
Isoleucine, di-TMMS	1308	-	0.56	-	-	2.16
γ-Aminobutyric acid, di-TMS	1310	-	0.15	-	-	0.46
Serine, tri-TMS	1370	-	0.17	0.16	-	1.67
Threonine, tri-TMS	1406	-	-	0.17	-	1.71
Pyroglutamic acid, di-TMS	1530	0.61	2.35	0.26	-	1.99
γ-Aminobutyric acid, tri-TMS	1541	-	0.31	0.16	-	-
Phenylalanine, di-TMS	1640	-	0.07	0.20	-	-
Glutamine, N,O,O-tri-TMS	1642	-	0.22	0.09	-	-
Asparagine, tri-O,O′,N-TMS	1690	-	0.61	0.38	-	-
Glutamine, N,‘O,O-tetra-TMS	1798	-	0.93	0.09	-	2.31
Triptophane, N,N′,O-tri-TMS	2236	-	0.19	-	-	0.29
**Monosaccharides and related compounds**		**54.95**	**40.00**	**61.23**	**45.39**	**52.11**
Threitol, tetra-TMS	1540	-	0.21	-	0.06	-
Arabinose, tetra-TMS	1649	0.07	-	0.07	0.09	-
Rhamnose, tetra-TMS	1660	0.12	-	0.06	0.07	-
Xylofuranose, tetra-TMS	1670	0.08	-	0.06	-	-
α-Ribofuranose, tetra-TMS	1678	trace	-	0.05	0.11	-
β-Ribofuranose, tetra-TMS	1680	0.06	-	0.11	0.09	-
Fucose, tetra-TMS	1699	-	-	0.11	0.07	-
Pentafuranose, TMS	1720	0.17	-	0.11	0.08	-
Xylopyranose, tetra-TMS	1736	0.23	-	0.56	0.94	-
Xylitol, penta-TMS	1745	-	-	0.06	-	-
Pentitol, penta-TMS	1757	0.25	-	0.29	0.45	-
Ribitol, penta-TMS	1762	-	-	0.05	-	0.38
Luxofuranose, tetra-TMS	1767	0.08	0.52	-	0.72	-
Arabinitol, penta-TMS	1776	0.24	0.34	-	-	-
β-Xylopyranose, tetra-TMS	1792	0.23	-	0.63	1.21	-
β-Xylofuranose, tetra-TMS	1799	0.25	-	0.26	0.30	-
α-Methylfuranoside, tetra-TMS	1812	0.48	-	0.12	0.47	-
Fucitol, penta-TMS	1817	-	-	-	0.08	-
Carbohydrate acid, TMS	1824	-	-	-	-	0.45
Carbohydrate acid, TMS	1843	-	-	-	-	0.61
Methyl-α-*D*-mannopyranoside, tetra-TMS	1828	-	-	-	068	-
Inositol, deoxy-, penta-TMS	1834	0.53	-	-	-	-
α-Fructofuranose, penta-TMS	1849	6.90	2.85	6.16	5.91	9.47
β-Fructofuranose, penta-TMS	1858	5.26	8.43	7.38	5.44	15.19
α-Galactofuranose, penta-TMS	1860	-	-	2.64	1.41	
Pinitol, penta-TMS	1872	-	-	0.88	-	
β-Fructopyranose, penta-TMS	1887	1.04	0.32	0.75	0.59	trace
β-Glucofuranose, penta-TMS	1892	1.22	0.56	1.00	0.95	1.33
α-Galactopyranose, penta-TMS	1899	-	0.51	-	-	-
Cyclohexanepentol, penta-TMS	1910	6.76	-	13.76	-	-
2-Amino-2-deoxyglucose, tetra-TMS	1912	1.17	-	-	-	-
α-Glucopyranose, penta-TMS	1932	11.34	9.50	8.24	9.57	11.78
β-Mannopyranose, penta-TMS	1943	0.52	-	-	0.52	-
β-Talopyranose, penta-TMS	1949	-	-	0.57	-	-
Mannitol, hexa-TMS	1970	0.41	0.41	0.13	0.28	trace
Glucitol, hexa-TMS	1976	-	0.14	0.87	trace	trace
Altitol, hexa-TMS	1984	-	-	0.29	0.41	-
Pinitol, isomer, penta-TMS	1996	1.04	-	-	-	-
*chiro*-Inositol, hexa-TMS	2000	3.62	trace	0.19	-	-
*scillo*-Inositol, hexa-TMS	2028	trace	-	-	-	0.46
β-Glucopyranose, penta-TMS	2032	7.86	10.85	8.79	8.04	12.78
Gluconic acid, hexa-TMS	2045	-	0.29	0.21	-	-
*myo*-Inositol, hexa-TMS	2131	2.25	1.77	2.68	2.05	trace
N-Acetylglucosamine, tetra-TMS	2143	0.14		-	0.19	-
Carbohydrate derivative, TMS	2226	-	-	-	-	0.55
Carbohydrate derivative, TMS	2373	-	-	-	-	0.23
**Polysaccharides and glycosides**		**16.98**	**32.65**	**13.98**	**30.72**	**9.73**
*myo*-Inositol phosphate	2260	-	0.26	-	-	-
2-O-Glycerol-α-*D*-galactopyranoside	2374	-	0.31	-	-	-
Glycosode, TMS	2439	-	-	-	-	0.41
Pyrocatechol β-*D*-glucopyranoside, penta-TMS	2491	-	0.47	-	-	-
Salicin, penta-TMS	2582	-	2.75	-	-	0.49
Disaccharide, TMS	2600	-	-	-	-	2.62
Arbutin, penta-TMS	2647	-	-	-	0.10	-
Disaccharide, TMS	2663	-	-	-	-	2.46
Xylobiose, hexa-TMS	2698	-	-	0.17	0.57	0.46
Sucrose, octa-TMS	2714	3.93	12.85	6.23	6.78	1.88
α-Maltose, octa-TMS	2746	-	-	0.15	0.37	1.92
Cellobiose, octa-TMS	2762	0.13	-	-	-	-
Turanose	2793	0.15	-	0.18	-	-
β-Maltose, octa-TMS	2803	-	0.30	0.14	0.08	-
Palatinose, octa-TMS	2818	-	-	0.18	-	-
Salidroside, penta-TMS	2832	-	-	-	1.86	-
Laminaribiose, octa-TMS, anomer 1	2864	-	-	0.48	-	-
Laminaribiose, octa-TMS, anomer 2	2891	-	-	1.35	-	-
Vanillic acid 4-β-glucoside	2937	-	-	-	0.43	-
Raffinose, undeca-TMS	3503	-	0.67	-	-	-
1-Kestose, undeca-TMS	3517	-	0.33	0.29	-	-
Erlose, undeca-TMS	3548	-	-	0.03	-	-
Syringin, penta-TMS	3143	-	0.20	-	-	-
Glycoside with 4-hydroxyphenylethanol moiety	3533	1.83	-	-	-	-
*epi*-Catechin-*O*-glycoside,octa-TMS? ***	3736	-	-	-	0.11	-
Kaempherol-3-β-*O*-galactoside, hepta-TMS	3742	-	-	0.51	-	-
Kaempherol-3-β-*O*-glucopyranoside, hepta-TMS	3755	-	-	0.46	-	-
Glycoside with quercetine moiety?, TMS	3806	-	-	0.16	-	-
Quercetin-3-α-*O*-galactoside, octa-TMS	3826	-	-	0.18	-	-
Quercetin-3-*O*-glucoside, octa-TMS	3836	-	-	0.08	-	-
Tremuloidin, tetra-TMS, isomer 1	3861	-	0.66	-	-	-
Catechin-7-*O*-glycoside,octa-TMS?	3866	-	-	-	7.18	-
Glycoside with 4-hydroxyphenylethanol moiety	3887	0.07	-	-	-	-
Tremuloidin, tetra-TMS, isomer 2	3897	-	0.54		-	-
Quercetin-3-α-*L*-arabinopyranoside, hepta-TMS	3918	0.40	-	-	1.43	-
Naringenin-7-*O*-glucoside, hexa-TMS?	3927	0.23	-	0.11	-	-
Glycoside with quercetine moiety, TMS	3931	-	-	-	23.08	-
β-Sitosterol-β-D-*O*-glucoside, tetra-TMS	>4000	1.41	-	1.27	0.87	-
**Aliphatic acids**		**8.56**	**5.53**	**4.26**	**4.74**	**6.72**
Lactic aci, di-TMS	1073	-	-	0.05	-	-
Glycolic acid, di-TMS	1083	0.04	-	0.03	-	-
Malonic, acid. Tri-TMS	1216	-	-	0.06	-	-
Succinic acid, di-TMS	1324	0.33	-	0.19	0.40	0.94
Glyceric acid, tri-TMS	1348	0.17	0.35	0.26	0.08	0.63
Fumaric acid, di-TMS	1355	trace	trace	0.05	-	0.58
Malic acid, tri-TMS	1510	4.85	2.65	1.04	3.29	1.39
2,3,4-Trihydroxybutyric acid, isomer 1, tetra-TMS	1575	-	0.91	trace	trace	0.37
2,3,4-Trihydroxybutyric acid, isomer 2, tetra-TMS	1597	-	0.14	0.92	0.12	0.62
L-Tartaric acid, tetra-TMS	1630	1.36	-	0.43	-	2.19
Azelaic acid, di-TMS	1808	0.10	-	-	-	-
Palmitic acid, mono-TMS	2052	0.42	0.48	0.44	0.27	trace
Linoleic acid, mono-TMS	2215	0.23	0.46	0.28	0.30	0.35
α-Linolenic & oleic acids, mono-TMS	2222	0.20	0.54	0.39	0.17	0.37
Stearic acid, mono-TMS	2249	0.14	trace	0.13	trace	trace
Docosanoic acid, mono-TMS	2646	-	-	-	0.10	-
**Aliphatic alcohols**		**1.07**	**1.24**	**2.14**	**0.96**	**1.75**
2,3-Butanediol, di-TMS, isomer 1	1042	-	-	0.05	-	-
2,3-Butanediol, di-TMS, isomer 2	1050	-	-	0.08	-	-
Glycerol, tri-TMS	1294	1.07	1.24	2.01	0.66	1.75
1-Dodecanol, mono-TMS	2558	-	-	-	0.30	-
**Aromatics**		**6.88**	**3.61**	**6.83**	**6.29**	**-**
Benzoic acid, mono-TMS	1246	-	0.37	-	-	-
Pyrocatechol, di-TMS	1321	-	0.12	-	-	-
4-Hydroxybenzaldehyde, mono-TMS	1373	0.05	-	-	-	-
Salicyl alcohol, di-TMS	1444	-	0.16	-	-	-
4-Hydroxyphenylethanol, di-TMS	1582	0.08	-	-	-	-
4-Hydroxybenzoic acid, di-TMS	1636	-	0.07	-	-	-
Protocatechuic acid, tri-TMS	1835	0.06	-	0.16	0.07	-
Methyl gallate. Tri-TMS	1920	1.40	-	3.16	-	-
*p*-Coumaric acid, di-TMS	1946	-	1.18	-	-	-
Gallic acid, tetra-TMS	1985	3.24	0.19	3.39	0.46	-
(*E*)-Ferulic acid, di-TMS	2101	-	0.32	-	-	-
*epi*-Catechin, penta-TMS	2909	-	-	-	0.79	-
*p*-Coumaroylquinate, penta-TMS, isomer 1	2909	-	-	0.26	-	-
Catechin, penta-TMS	2940	0.78	0.27	-	3.63	-
Apigenin, 7,4′-di-TMS	3082	-	0.45	-	-	-
Kaempherol, tetra-TMS	3115	-	-	0.07	-	-
*p*-Coumaroylquinate, penta-TMS, isomer 2	3123	0.21	-	0.07	0.19	-
Apigenin, tri-TMS	3158	-	0.17	-	-	-
Chlorogenic acid, hexa-TMS	3186	0.24	0.30	-	0.43	-
Quercetin, penta-TMS	3210	0.09	-	-	0.60	-
Ellagic acid, tetra-TMS	3335	0.45	-	0.05	-	-
Procyanidin B1, deca-TMS	>4000	-	-	-	0.75	-
Cryptochlorogenic acid, hexa-TMS	3256	0.09	-	-	-	-
Neochlorogenic acid, hexa-TMS	3272	0.19	-	-	-	-
**Other compounds**		**3.25**	**2.35**	**1.42**	**4.88**	**6.17**
Phosphoric acid, tri-TMS	1290	1.31	0.46	0.62	0.15	1.25
2-Pyrrolidone-5-carboxylic acid, mono-TMS	1505	0.51	-	-	-	-
α-Glycerophosphoric acid, tetra-TMS	1797	0.21	0.10	0.13	-	0.50
Dihydroxyacetone, dimer, tetra-TMS	1826	-	0.15	-	-	-
Quiniv acid	1902	-	-	-	-	0.52
Ascorbic acid, tetra-TMS	1981	0.35	-	-	-	-
Uridine, tri-TMS	2468	-	-	0.13	0.08	-
Adenosin riboside, tetra-TMS	2672	-	-	0.17	-	-
*n*-Pentacosane	2500	-	-	-	0.14	2.62
β-Sitosterol, mono-TMS	3349	0.17	0.24	0.37	0.37	1.35
Esters (docosanoate?)	>4000	-	-	-	2.98	-
**NN**		**7.70**	**7.21**	**7.19**	**7.02**	**3.18**
**Peak number**		**97**	**97**	**133**	**118**	**67**

* not found; ** below 0.01% of TIC; *** the identification of the corresponding compound is considered preliminary.

## Data Availability

Data are contained within the article and Appendix A.

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
