# Peer review of "Chemical Composition of Volatile and Extractive Organic Compounds in the Inflorescence Litter of Five Species of Woody Plants"

_plants, 2024, doi:10.3390/plants13131829_

Round 1
Reviewer 1 Report
Comments and Suggestions for Authors
This study analyzed chemical compositions of volatile and extractive organic compounds in the inflorescence litter of four species of forest-forming trees. In general, the experiments were well-performed and the manuscript was well-written. It may be published pending some minor revisions.
1) In the title, "some" should be replaced by " four species of".
2) Not all abbreviations have been introduced with full names. For example, "HS-SPME/GC-MS" in Line 96, "BSTFA/TMCS" in Line 128 and "NN" in Table 2.
3) Pay attention to the subscript, e.g. "CO2" in Line 9, "C20H32" in Table 1 and "MgSO4" in Line 126. All Latin names should be in italics, e.g. in Lines 164, 165 and Legends of Table 1 and Table 2. In Table 1, "3Methyloctan-2-ol?", what does the question mark mean?
4) Although 1-octen-3-ol is known as “mushroom alcohol”, it can be synthesized in some plant species. To prove the conclusion that inflorescence-derived VOCs are a mixture of components of plant and microbial origin, plant VOCs and microbial VOCs indentified in this study should be introduced and discussed separately.
5) All the data are expressed as the mean ± standard error of the mean (SEM) from at least three independent biological repeats, even for the percentage data. And groups of multiple comparisons should be marked within each species.
Author Response
1. In the title, "some" should be replaced by " four species of". |
The reviewer's suggestion has been accepted and a corresponding change has been made. |
2) Not all abbreviations have been introduced with full names. For example, "HS-SPME/GC-MS" in Line 96, "BSTFA/TMCS" in Line 128 and "NN" in Table 2. |
The appropriate changes have been made (lines 76-77 and 106-107). |
3) Pay attention to the subscript, e.g. "CO2" in Line 9, "C20H32" in Table 1 and "MgSO4" in Line 126. All Latin names should be in italics, e.g. in Lines 164, 165 and Legends of Table 1 and Table 2. In Table 1, "3Methyloctan-2-ol?", what does the question mark mean?
|
The appropriate changes have been made. The question mark indicates that the identification of the corresponding compound is considered preliminary or presumptive. The corresponding explanation is given below Tables 1 and 2. |
4) Although 1-octen-3-ol is known as “mushroom alcohol”, it can be synthesized in some plant species. To prove the conclusion that inflorescence-derived VOCs are a mixture of components of plant and microbial origin, plant VOCs and microbial VOCs indentified in this study should be introduced and discussed separately. |
As far as I know, the "mushroom alcohol" 1-octen-3-ol is found in plants infected with some types of fungi. This confirms its "mushroom" origin. |
All the data are expressed as the mean ± standard error of the mean (SEM) from at least three independent biological repeats, even for the percentage data. And groups of multiple comparisons should be marked within each species.
|
We do not quite understand this remark (or statement?). The paper provides data on the content of components that cannot be considered quantitative. To obtain quantitative data (it is for them that the definition and presentation of SEM values ​​makes sense) it would be necessary to calibrate the mass-spectral detector for EVERY component being recorded, which is completely impossible, if only because ALL standard substances are missing, not to mention the enormous labor and time costs. However, in lines 123-127 we provide an additional explanation of our approach to assessing the precision of the method. It seems to us that this is quite sufficient. |

Reviewer 2 Report
Comments and Suggestions for Authors
In a manuscript presented by Valery Isidorov and Jolanta MasÅ‚owiecka, it was hypothesized that the release of volatiles from spent inflorescences continues even after they have fallen and thus serves as a source of highly reactive VOCs during the spring–summer transition period. The purpose of this work was to study the chemical composition of volatile emissions of fallen inflorescences from four tree species, i.e. silver birch (Betula pendula Roth.), European hornbeam (Carpinus betulus L.), black alder (Alnus glutinosa Gaertn.) and aspen (Populus tremula L.) and determining the composition of the extractive compounds contained them.
In this review, I offer a few suggestions as to where certain points can be elaborated upon or revised in the manuscript
Detailed comments:
1. Lines 82-84; 181-182; 211; 227: Latin names are written in italics.
Section 2.2. Requires additions and explanations. Below are the tips (point 2-8).
2. Line 85: Please provide the area of ​​“Las Zwierzyniecki” Nature Reserve near Bialystok, Poland
3. Line 85: Please add geographic coordinates for this area.
4. Please add a brief description of this area in terms of species composition. Justifying why these 4 species were selected for research. Do they dominate the flora of this area, country (Poland) or Europe?
5. Please add information on how the purity of the analyzed plant material was ensured in field conditions? How can we be sure that there were no flowers (contaminants) of another species in the collected inflorescences?
6. Were the inflorescences of some of the analyzed species that were taken as test samples already in the decomposition phase? The flowering time of the studied species varies. However, the authors state in section 2.2 that inflorescences were collected for all species in early May 2022.
7. How many samples were there for each species? Please provide the number of samples with details separately for freshly collected and incubated inflorescences.
8. How long was the mulch bags exposed to natural conditions? Please move this information from section 3.1 (line 157).
9. Line 157: „The composition of volatile emissions was determined for freshly fallen inflorescences and after two weeks of exposure to natural conditions” - This information should be moved to section 2.2.
10. Lines 179-182: Table 1 contains data for 2 species (Acer platanoides; Salix fragilis) that were not the subject of the study. Please add a reference to these results. Tables 1, S1 and S2 do not show results for Alnus glutinosa. Tables S1 and S2 present the test results for A. platanoides, this species was not the subject of the study.
11. Conclusions: Specifically how and where these results can be used. Furthermore, please consider that the conclusion is intended to help the reader understand why your research should matter to them. A conclusion is not merely a summary of the main results but a synthesis of key points and where you recommend new areas for future research.
12. Reference Lines 291-399: References not prepared in accordance with the requirements journal Plants (Plants | Instructions for Authors (mdpi.com). For example:
Author 1, A.B.; Author 2, C.D. Title of the article. Abbreviated Journal Name Year, Volume, page range.
Author Response
1. Lines 82-84; 181-182; 211; 227: Latin names are written in italics. Section 2.2. Requires additions and explanations. Below are the tips (point 2-8). |
The comment has been taken into account. |
2. 2. Line 85: Please provide the area of ​​“Las Zwierzyniecki” Nature Reserve near Bialystok, Poland 3. 3. Line 85: Please add geographic coordinates for this area.. . 4. 4. Please add a brief description of this area in terms of species composition. Justifying why these species were selected for research. Do they dominate the flora of this area, country (Poland) or Europe? |
The relevant data has been added to this section (lines 86-91). |
4 5.Please add information on how the purity of the analyzed plant material was ensured in field conditions? How can we be sure that there were no flowers (contaminants) of another species in the collected inflorescences? |
See lines 92 and 99. |
. 6. Were the inflorescences of some of the analyzed species that were taken as test samples already in the decomposition phase? The flowering time of the studied species varies. However, the authors state in section 2.2 that inflorescences were collected for all species in early May 2022.
|
The flowering time of all the above woody plant species in spring 2022 fell on the same time. We tried to collect freshly fallen inflorescences, but, of course, it cannot be ruled out that some of them were affected by decomposition to some extent. Moreover, it cannot be ruled out that it begins when the dead inflorescences are still on the branches of the «mother» trees. We think that there is no need to distract readers' attention from the main content of the article. |
7. How many samples were there for each species? Please provide the number of samples with details separately for freshly collected and incubated inflorescences. |
For each type of litter, one large sample was collected and divided into parts. One of them was used to determine the VOCs and extractives of the "fresh" litter, and the second was placed in litter-bags to record changes in composition during incubation. It was written about this (see lines 94-96 and 103-106). |
8. How long was the mulch bags exposed to natural conditions? Please move this information from section 3.1 (line 157). |
The duration of this stage of the experiment was 14 days. (added to section 2.2, line 101) |
. 9. Line 157: „The composition of volatile emissions was determined for freshly fallen inflorescences and after two weeks of exposure to natural conditions” - This information should be moved to section 2.2.
|
We don't see much need for this, given what we wrote at the very beginning of section 2.3 (lines 104-105). |
10. Lines 179-182: Table 1 contains data for 2 species (Acer platanoides; Salix fragilis) that were not the subject of the study. Please add a reference to these results. Tables 1, S1 and S2 do not show results for Alnus glutinosa. Tables S1 and S2 present the test results for A. platanoides, this species was not the subject of the study. |
Indeed, this is our unfortunate omission. The study covered 5 types of litter: hornbeam, maple, aspen, birch and willow. The corresponding corrections have been made in section 2.2 and in the tables. Data on the composition of VOCs of black alder (A. glutinosa) are not included in this report and their mention is erroneous. |
11. Conclusions: Specifically how and where these results can be used. Furthermore, please consider that the conclusion is intended to help the reader understand why your research should matter to them. A conclusion is not merely a summary of the main results but a synthesis of key points and where you recommend new areas for future research |
It seems to us that the conclusion contains not so much a summary of the results obtained, as an indication of the importance of continuing studies of the composition of both volatile and extractive components of the inflorescence litter of other plant species, the number of which is enormous. This is important both in the atmospheric and biogeospheric terms. |
12. Reference Lines 291-399: References not prepared in accordance with the requirements journal Plants (Plants | Instructions for Authors (mdpi.com). For example: Author 1, A.B.; Author 2, C.D. Title of the article. Abbreviated Journal Name Year, Volume, page range.
|
The comment has been taken into account. |

Round 2
Reviewer 2 Report
Comments and Suggestions for Authors
The authors' responses refer to specific lines. Please attach the appropriate file with numbered lines, which will enable another review.
Author Response
Reviewer 2 |
|
1. Lines 82-84; 181-182; 211; 227: Latin names are written in italics. Section 2.2. Requires additions and explanations. Below are the tips (point 2-8). |
The comment has been taken into account. |
2. 2. Line 85: Please provide the area of ​​“Las Zwierzyniecki” Nature Reserve near Bialystok, Poland 3. 3. Line 85: Please add geographic coordinates for this area.. . 4. 4. Please add a brief description of this area in terms of species composition. Justifying why these species were selected for research. Do they dominate the flora of this area, country (Poland) or Europe? |
The relevant data has been added to this section (lines 86-91). |
4 5.Please add information on how the purity of the analyzed plant material was ensured in field conditions? How can we be sure that there were no flowers (contaminants) of another species in the collected inflorescences? |
See lines 92 and 99. |
. 6. Were the inflorescences of some of the analyzed species that were taken as test samples already in the decomposition phase? The flowering time of the studied species varies. However, the authors state in section 2.2 that inflorescences were collected for all species in early May 2022.
|
The flowering time of all the above woody plant species in spring 2022 fell on the same time. We tried to collect freshly fallen inflorescences, but, of course, it cannot be ruled out that some of them were affected by decomposition to some extent. Moreover, it cannot be ruled out that it begins when the dead inflorescences are still on the branches of the «mother» trees. We think that there is no need to distract readers' attention from the main content of the article. |
7. How many samples were there for each species? Please provide the number of samples with details separately for freshly collected and incubated inflorescences. |
For each type of litter, one large sample was collected and divided into parts. One of them was used to determine the VOCs and extractives of the "fresh" litter, and the second was placed in litter-bags to record changes in composition during incubation. It was written about this (see lines 94-96 and 103-106). |
8. How long was the mulch bags exposed to natural conditions? Please move this information from section 3.1 (line 157). |
The duration of this stage of the experiment was 14 days. (added to section 2.2, line 101) |
. 9. Line 157: „The composition of volatile emissions was determined for freshly fallen inflorescences and after two weeks of exposure to natural conditions” - This information should be moved to section 2.2.
|
We don't see much need for this, given what we wrote at the very beginning of section 2.3 (lines 104-105). |
10. Lines 179-182: Table 1 contains data for 2 species (Acer platanoides; Salix fragilis) that were not the subject of the study. Please add a reference to these results. Tables 1, S1 and S2 do not show results for Alnus glutinosa. Tables S1 and S2 present the test results for A. platanoides, this species was not the subject of the study. |
Indeed, this is our unfortunate omission. The study covered 5 types of litter: hornbeam, maple, aspen, birch and willow. The corresponding corrections have been made in section 2.2 and in the tables. Data on the composition of VOCs of black alder (A. glutinosa) are not included in this report and their mention is erroneous. |
11. Conclusions: Specifically how and where these results can be used. Furthermore, please consider that the conclusion is intended to help the reader understand why your research should matter to them. A conclusion is not merely a summary of the main results but a synthesis of key points and where you recommend new areas for future research |
It seems to us that the conclusion contains not so much a summary of the results obtained, as an indication of the importance of continuing studies of the composition of both volatile and extractive components of the inflorescence litter of other plant species, the number of which is enormous. This is important both in the atmospheric and biogeospheric terms. |
12. Reference Lines 291-399: References not prepared in accordance with the requirements journal Plants (Plants | Instructions for Authors (mdpi.com). For example: Author 1, A.B.; Author 2, C.D. Title of the article. Abbreviated Journal Name Year, Volume, page range.
|
The comment has been taken into account. |

Round 3
Reviewer 2 Report
Comments and Suggestions for Authors
The comments are included in the attached file.
